

# Morphological and olfactory tree traits influence the susceptibility and suitability of the apple species *Malus domestica* and *M. sylvestris* to the florivorous weevil *Anthonomus pomorum* (Coleoptera: Curculionidae)

Benjamin Henneberg[1], Torsten Meiners[2], Karsten Mody[3] and Elisabeth Obermaier[1]

[1] Ecological-Botanical Garden of the University of Bayreuth, University of Bayreuth, Bayreuth, Germany
[2] Julius Kühn-Institut, Federal Research Centre for Cultivated Plants, Ecological Chemistry, Plant Analysis and Stored Product Protection, Berlin, Germany
[3] Department of Applied Ecology, Hochschule Geisenheim University, Geisenheim, Germany

## ABSTRACT

The florivorous apple blossom weevil, *Anthonomus pomorum* (Coleoptera: Curculionidae), is the most economically relevant insect pest of European apple orchards in early spring. Neither efficient monitoring nor ecologically sustainable management of this insect pest has yet been implemented. To identify heritable traits of apple trees that might influence the host selection of *A. pomorum*, we compared the susceptibility of apple tree species using infestation rates of the domesticated apple, *Malus domestica* (Rosaceae: Pyreae), and the European crab apple, *M. sylvestris*. We evaluated the suitability of the two apple species for *A. pomorum* by quantifying the mass of weevil offspring. Because volatile organic compounds (VOCs) emitted from flower buds of the domesticated apple have previously been suggested to mediate female weevil preference via olfactory cues, we conducted bioassay experiments with blossom buds of both apple species to explore the olfactory preference of adult weevils and, furthermore, identified the headspace VOCs of blossom buds of both apple species through GC-MS analysis. The infestation analysis showed that *A. pomorum* infested the native European crab apple more prevalently than the domesticated apple, which originated from Central Asia. The European crab apple also appeared to be better suited for weevil larval development than the domesticated apple, as weevils emerging from *M. sylvestris* had a higher body mass than those emerging from *M. domestica*. These field observations were supported by olfactory bioassays, which showed that *A. pomorum* significantly preferred the odor of *M. sylvestris* buds compared to the odor of *M. domestica* buds. The analysis of headspace VOCs indicated differences in the blossom bud volatiles separating several *M. domestica* individuals from *M. sylvestris* individuals. This knowledge might be employed in further studies to repel *A. pomorum* from *M. domestica* blossom buds.

Corresponding author
Benjamin Henneberg,
benjamin1.henneberg@uni-bayreuth.de

## INTRODUCTION

The apple blossom weevil, *Anthonomus pomorum* L. (Coleoptera: Curculionidae) is one of the economically most important insect pest herbivores of apple trees (*Collatz & Dorn, 2013*; *Knuff, Obermaier & Mody, 2017*). It is a widespread univoltine insect pest in European apple orchards directly reducing the fruit harvest of the domesticated apple, *Malus domestica* Borkh. (Rosaceae: Pyreae), by infesting the unopened blossom buds early in spring (*Duan, Weber & Dorn, 1998*; *Toepfer, Gu & Dorn, 1999*; *Daniel, Tschabold & Wyss, 2005*; *Žďárek et al., 2013*). The apple blossom weevil used to be a minor pest in European apple orchards, as the use of broad-spectrum insecticides provided adequate control. However, in recent decades the economic importance of this pest insect has increased (*Toepfer, Gu & Dorn, 1999*). It is now considered the most damaging pest weevil of apple throughout Europe (*Mody, Collatz & Dorn, 2015*; *Zabrodina et al., 2020*). In some European countries, population densities of the apple blossom weevil have exceeded economic thresholds due to changes in pest management (*Toepfer, Gu & Dorn, 1999*; *Sipos & Marko, 2014*). The percentage of blossoms infested and destroyed by *A. pomorum* has reached 60–90 % as was reported for Lithuania (*Toepfer, Gu & Dorn, 1999*), or even up to 85–100 % in organic apple orchards in Slovenia (*Bajec et al., 2013*). *Zabrodina et al. (2020)* call the apple blossom weevil "one of the most harmful insects in horticulture" and report yield losses for Russia and the Ukraine of 50–100 %. The increasing pest status of *A. pomorum* poses a serious threat to organic apple orchards and integrated pest management (IPM) systems currently in use in European apple orchards (*Miñarro & García, 2018*; *Shaw, Nagy & Fountain, 2021*).

Although the biology and population ecology of *A. pomorum* is relatively well documented, neither efficient monitoring nor management programs for this pest insect have yet been developed, let alone implemented (*Toepfer, Gu & Dorn, 1999*). There are two main monitoring methods: limb jarring (which is inconvenient, *Toepfer, Gu & Dorn, 1999*) and shelter traps (*Hausmann, Samietz & Dorn, 2004a*; *Hausmann, Samietz & Dorn, 2004b*). Shelter traps are more preferred as they exploit the weevils' preference for warm microclimates early in the year and may serve as a useful tool for assessing the number of weevils that colonize the tree by crawling up the trunk (*Hausmann, Samietz & Dorn, 2004b*). However, flight has been indicated as the most important mode of tree colonization in *A. pomorum* (*Toepfer, Gu & Dorn, 2002*). As flight-guiding pheromones are still unknown for this pest insect (*Dorn & Piñero, 2009*), flight traps baited with attractive host-plant odors might constitute an efficient monitoring tool (*Natale et al., 2003*). Flight traps baited with attractive odors may present a useful component of novel strategies for an ecologically sustainable apple production since they do not rely on broad-spectrum insecticides. To develop these strategies, it is of great importance to gain more insight in the underlying mechanism(s) of host tree selection by *A. pomorum* (*Knuff, Obermaier & Mody, 2017*).

Many studies have been conducted on the biology of *A. pomorum* (*Čtvrtečka & Žďárek, 1992*; *Duan et al., 1996*; *Duan, Weber & Dorn, 1998*; *Toepfer, Gu & Dorn, 1999*; *Kalinová et al., 2000*; *Toepfer, Gu & Dorn, 2000*; *Toepfer, Gu & Dorn, 2002*; *Hausmann, Samietz & Dorn, 2004a*; *Hausmann, Samietz & Dorn, 2004c*; *Hausmann, Samietz & Dorn, 2005*;

*Piskorski & Dorn, 2010*; *Collatz & Dorn, 2013*; *Knuff, Obermaier & Mody, 2017*), but still little is known about the cues that female weevils use when colonizing apple orchards in early spring. Following colonization from adjacent forests (*Hausmann, Samietz & Dorn, 2004a*; *Dorn & Piñero, 2009*), females lay their eggs into closed blossom buds (*Mody, Collatz & Dorn, 2015*). Differences in infestation between apple species or among cultivars of *M. domestica* are probably strongly mediated by preference behavior of female weevils searching for suitable host trees for oviposition (*Hogmire & Miller, 2005*; *Mody, Collatz & Dorn, 2015*). Therefore, an identification of apple tree characteristics that guide female *A. pomorum* in search of their host would be a first step for breeding less susceptible apple cultivars (*Knuff, Obermaier & Mody, 2017*), or to develop effective and ecologically sustainable monitoring and management tools. Because *A. pomorum* infests apple blossom buds at certain developmental stages (*Toepfer, Gu & Dorn, 2002*), the traits of blossom buds are particularly promising for breeding apple cultivars that are less attractive and consequently less susceptible to the apple blossom weevil (*Mody, Collatz & Dorn, 2015*) or to use these traits as a screening mechanism for newly developed cultivars. To identify heritable traits of apple trees that might influence host selection of female *A. pomorum*, it may be informative to study oviposition preferences for different *Malus* species, as differences are assumedly more pronounced between species than among cultivars of the same species (*Knuff, Obermaier & Mody, 2017*).

As different plant species or individuals vary in their suitability as host for the offspring of a given herbivore, females should have evolved an ability to identify the host plant best suited for their offspring (*Jaenike, 1978*; *Thompson, 1988*; *Craig, Itami & Price, 1989*; *Gripenberg et al., 2010*). Preference of oviposition in females determines host plant infestation by their offspring in many insect herbivores (*Gripenberg et al., 2010*; *Mody, Collatz & Dorn, 2015*), leading to positive preference-performance relationships between female preference and offspring performance (*Mody, Collatz & Dorn, 2015*).

The apple blossom weevil, as stated above, is an important pest insect of domesticated apple trees in Europe, but it is also likely to infest other species of the genus *Malus*, like the European crab apple, *M. sylvestris* (L.) Mill. So far, there is only one published study on the ecology of the apple blossom weevil on *M. sylvestris* trees (*Knuff, Obermaier & Mody, 2017*); further studies that assess the specific entomofauna of this wild apple species are still lacking (*Mody, 2013*). Whereas the origin of the domesticated apple lies in the Tian Shan region of Central Asia (*Velasco et al., 2010*; *Cornille et al., 2012*), the European crab apple is the only wild apple species that is indigenous to Central Europe (*Robinson, Harris & Juniper, 2001*; *Reim, Höltken & Höfer, 2013*). It is a rare and endangered species (Red List Bavaria: category 3) (*LfU, 2003*) that grows in open forests, forest edges and in hedgerows (*Stephan, Wagner & Kleinschmit, 2003*; *Aas, 2013*; *Knuff, Obermaier & Mody, 2017*). However, *M. sylvestris* easily hybridizes with *M. domestica* (*Reim, Höltken & Höfer, 2013*), and bidirectional gene flow between the two apple species in Europe has resulted in the current *M. domestica* being genetically more closely related to *M. sylvestris* than to its progenitors from Central Asia (*Cornille et al., 2012*).

This study deals with the question of whether the apple blossom weevil shows specific host selection patterns for the apple species *M. domestica* and *M. sylvestris*. A better

understanding of the observed patterns could lead to an identification of apple tree characteristics that may be used for breeding less susceptible cultivars or, in the case of volatile organic compounds (VOCs) emitted from the blossom buds, developing effective and sustainable monitoring tools. Plant resistance to arthropods is often mediated by morphological and phenological plant traits (*Smith, 2005*; *Miñarro & Dapena, 2008*; *Smith & Clement, 2012*; *Knuff, Obermaier & Mody, 2017*), and most knowledge on the expression of plant resistance is available for herbivores feeding on leaves (*Kessler & Baldwin, 2002*; *Dicke & Hilker, 2003*; *Van Dam, 2009*; *Mody, Collatz & Dorn, 2015*). Much less is known about whether plant resistance is expressed similarly with regard to flower-feeders (*McCall & Irwin, 2006*; *Oguro & Sakai, 2014*).

As mentioned above, a particularly promising trait to mediate the preference of female *A. pomorum* searching for suitable host trees, besides physical plant characteristics, is the spectrum of VOCs emitted from the blossom buds (*Kalinová et al., 2000*; *Piskorski & Dorn, 2010*; *Mody, Collatz & Dorn, 2015*). It has been shown through chemical analyses of plant headspace VOCs that were combined with behavioral bioassays that host-plant odors play an important role in host location for lepidopteran pest insects, like the oriental fruit moth *Cydia molesta* Busck (*Natale et al., 2003*), or the codling moth *C. pomonella* (*Vallat & Dorn, 2005*; *Piskorski & Dorn, 2010*). Studies also showed for many species of curculionids that they regularly rely on olfactory cues for host detection, *e.g.*, the pine weevil *Hylobius abietis* L. (*Kännaste et al., 2009*), the strawberry blossom weevil *Anthonomus rubi* Herbst (*Bichão et al., 2005*), or the boll weevil *Anthonomus grandis* Boheman (*Minyard et al., 1969*; *Dickens, 1989*), the latter two being close relatives of the apple blossom weevil. However, despite an identified blend of VOCs released by blossom buds of prebloom *M. domestica* trees (*Piskorski & Dorn, 2010*), knowledge on VOCs from apple blossom buds released before bloom in the time of host-tree selection of apple blossom weevils is still scarce. Further identification of such VOCs that control the oviposition behavior of female *A. pomorum* may be of great practical significance aside from future apple breeding programs: if effective attractants of the apple blossom weevil are found, synthetically produced kairomones could be used in traps in apple orchards to monitor or even control weevil populations (*Kalinová et al., 2000*; *Piskorski & Dorn, 2010*).

The research questions that are addressed in this study are therefore (i) does *A. pomorum* show certain host selection patterns among the two *Malus* species, measured as infestation rate, indicating an oviposition preference of females? (ii) Does performance of offspring, indicated by weevil mass of newly hatched imagines, differ among the two *Malus* species, and if so, how does it relate to preference of females? (iii) How do adult overwintered weevils behaviorally react to the odor of blossom buds of the two apple species in olfactory bioassays? (iv) Does the spectrum of VOCs emitted from the blossom buds differ among the two *Malus* species, and which compounds can be identified through sampling of headspace VOCs?
## MATERIALS & METHODS

### Study sites

The study was conducted in 2017 in the Ecological-Botanical Garden of the University of Bayreuth (49°55′N, 11°35′E, elevation: 355 m a. s. l.) and in the surroundings of Bayreuth. Two apple species were investigated for their infestation by the apple blossom weevil *A. pomorum*: the domesticated apple, *M. domestica*, and the European crap apple, *M. sylvestris*. Studied *M. domestica* trees grew in the botanical garden on a meadow with scattered fruit trees in rows with a spacing of 8 m x 6 m. The orchard contained 98 high stem apple trees of 81 different cultivars and trees of several *Pyrus* and *Prunus* species planted in between the apple trees. The lower part of the meadow was surrounded by a hedgerow in the south, grassland with some shrubs in the west, beds with useful plants in the north, and further fruit trees in the east. For this study, ten *M. domestica* trees of ten different cultivars planted between 1998 and 2008 were randomly selected (Table S1) so that they were distributed over the whole area of the fruit orchard. Neither pesticides nor fungicides were applied to the apple trees during the study period.

The *M. sylvestris* trees used in this study comprised two individuals (genotypes) that grew inside the botanical garden, and eight individuals that grew in hedgerows, riverbanks, or forest margins in the surroundings of Bayreuth (Table S1).

For the sampling of headspace VOCs, smaller potted trees of *M. domestica* and *M. sylvestris* were used that we were able to move to the laboratory (Table S2).

### Infestation assessment and sampling of infested blossom buds

Female weevils lay their eggs in blossom buds of certain developmental stages, namely the bud stages 56 and 57 according to BBCH (*Toepfer, Gu & Dorn, 2002*), shortly before the opening of the flowers. Therefore, phenology of the blossom buds was assessed for both tree species following BBCH (*Meier, 2001*; *Meier et al., 2009*) at weekly intervals. Infestation by *A. pomorum* was assessed shortly after full flowering from mid-May to end of May by counting the number of infested (recognizable as unopened blossom buds with brownish, dead petals forming a hollow cavity, so-called "capped blossoms") and non-infested blossom buds of ten blossom clusters on five branches, with two clusters per branch. Branches had been randomly selected in a way that they were evenly distributed over the treetop (*Knuff, Obermaier & Mody, 2017*).

After infestation assessment, as many capped blossoms as possible were collected from each tree (666 in total; per tree: 33.3 $\pm$ 15.8 (mean $\pm$ SD)) (Table 1). Capped blossoms were kept in plastic containers with insect-proof gauze in the lids allowing for sufficient ventilation. The containers were equipped with paper towels for moisture absorption and kept inside the laboratory at approximately 22 °C until the insects had hatched.

### Weevil characteristics

Plastic containers containing the capped blossoms were examined for the presence of freshly emerged weevils at 24h-intervals. After emergence, weevils were immediately removed from the containers and deep-frozen and stored in Eppendorf tubes at −16 °C (*Mody, Collatz & Dorn, 2015*; *Knuff, Obermaier & Mody, 2017*). Emergence was

**Table 1  Numbers of infested capped blossoms that were collected from *Malus domestica* and *M. sylvestris* trees and numbers of weevils that emerged from the capped blossoms.** Numbers of capped blossoms per tree: 33.3 ± 15.8 (mean ± SD). Discrepancies between number of collected blossoms and total number of emerged weevils are due to parasitoids and weevils that could not complete their development.

| Tree species | Number of collected capped blossoms | Total number of emerged weevils (males/females) |
|---|---|---|
| *M. domestica* | 282 | 139 (74/65) |
| *M. sylvestris* | 384 | 281 (132/149) |

monitored until no more insects had emerged for seven consecutive days. The sex of emerged weevils was determined based on morphology of the last dorsal abdominal plate (*Duan et al., 1999*). All weevils were then dried at 40 °C to mass constancy and weighed (Ohaus Explorer EX423M high-precision scale, Ohaus Europe GmbH, Greifensee, Switzerland). Weevil weight was calculated as the average weight of all individuals per tree and sex to analyze the tree species' suitability for weevil development.

## Olfactory bioassays

As many post-diapause *A. pomorum* imagines as possible were collected from *M. domestica* trees by limb-jarring during their tree colonization period in the botanical garden, from mid-March to mid-April. Weevils were kept in plastic vials (4.5 cm diameter, 10 cm height; approximately 10 individuals per vial) for up to one week. They were kept under controlled conditions in a climate chamber at 12/12 h day/night, 12°/8 °C, and 70% relative humidity (RH). Weevils were offered moist apple leaves ad libitum as food and shelter (*Piskorski & Dorn, 2010*).

Olfactory bioassays were performed as still-air dual-choice experiments in plastic Petri dishes as described by *Prokopy, Cooley & Phelan (1995)* (Fig. S1). Two holes (10 mm diameter, 70 mm apart) were bored through the lid of a Petri dish (14 cm diameter, 1.5 cm high). A polyethylene micropipet tip (10 mm diameter at the base, 20 mm high, and seven mm diameter at the top after cutting off the tip) was fitted snugly into each hole so that the base was flush with the lid of the Petri dish and the tip protruded above the lid. A 100-ml transparent polypropylene cup was centered over each pipet tip, enclosing it (Fig. S1). One or both of the cups contained a treatment in the form of an apple twig of approximately five cm length carrying 5 ± 1 blossom buds that were in the suitable stage for oviposition (*Toepfer, Gu & Dorn, 2002*). The other cup contained either a twig of the second apple species or an appropriate control in the form of a model of an apple twig made out of wire. Cultivars of *M. domestica* or genotypes of *M. sylvestris* that were used in the form of apple twigs for olfactory bioassays represented a subset of the individuals used for the infestation analysis, namely the cultivars "Roter Boskoop", "Blauacher Wädenswil", and "James Grieve", as well as the genotypes "Destuben", "Schlehenmühle", and "Lochau 2". To prevent the apple twigs from drying out and subsequently changing their odor profile, each twig was placed in an Eppendorf tube containing water. Thereby, the cut surface

of each apple twig was covered with water to avoid that the inflicted mechanical damage altered the composition of the volatile blend (*Collatz & Dorn, 2013*). Conditions in the test chamber were kept constant at 16 °C and 70% RH.

Prior to the start of bioassays, the weevils were deprived of food for 24 h and allowed to acclimatize to the conditions in the test chamber for 90 min. At 3.5 h after the onset of photophase, a single weevil was released on the bottom of each Petri dish, which was covered with circular filter paper and tightly sealed with parafilm (Pechiney Plastic Packaging, Chicago, IL). Bioassays were run for 180 min, after which the choice of each weevil was recorded (*Collatz & Dorn, 2013*). Each weevil was only used once in bioassays. New apple twigs and a clean Petri dish were used for each bioassay. The sex of weevils that were tested in bioassays was not recorded, because odor preference had been shown to not differ between male and female apple blossom weevils (*Collatz & Dorn, 2013*).

The behavioral response of apple blossom weevils to the odor of *M. domestica* twigs with blossom buds was assessed against controls ($n = 50$), as well as the behavioral response to twigs of *M. sylvestris* against controls ($n = 50$). Twigs of *M. domestica* and *M. sylvestris* were also compared directly in bioassays, so that weevils could choose between the two host-plant species ($n = 50$). To assure that the controls did not have any influence on the behavioral response of weevils, neither attracting nor repellent, the controls were also tested against blanks ($n = 30$).

## Sampling of VOCs

The headspace VOCs of apple blossom buds were sampled in late April, when blossom buds were in bud stages 56 and 57 according to BBCH (*Meier, 2001*), suitable for oviposition by apple blossom weevils (*Toepfer, Gu & Dorn, 2002*). It has been found that apple blossom weevils show their highest levels of flight activity at bud stages 56 and 57 and that oviposition is restricted to these bud stages (*Toepfer, Gu & Dorn, 2002*; *Zabrodina et al., 2020*). Ten potted individuals of six different cultivars of *M. domestica* were sampled, as well as eight potted individuals of four different genotypes of *M. sylvestris* (Table S2). Originally, ten individuals of each apple species were sampled, but two *M. sylvestris* individuals had to be removed from the study due to being identified as hybrids. Each tree was sampled three times on the same day without precipitation, yielding 30 samples of *M. domestica* and 24 samples of *M. sylvestris*. The sampling of VOCs was conducted in the laboratory at 18 °C, 60% RH, using headspace sorptive extraction (SBSE) (*Bicchi et al., 2000*). Twigs carrying $10 \pm 1$ blossom buds were enclosed in a PET oven bag (Toppits Bratschlauch, Cofresco GmbH, Minden, Germany) together with a 10-mm Gerstel©Twister (Gerstel, Mühlheim, Germany) that consists of a magnetic stir bar enclosed in glass and coated with 0.5 mm polydimethylsiloxane (PDMS) as sorbent. Each Twister was held by a magnet from the outside of the PET oven bag. Sampling of headspace VOCs lasted 6 h.

## Analysis of VOCs

The VOCs were thermally desorbed from the PDMS twisters with a Gerstel©Twister Desorption Unit (TDU), and the chemical composition of headspace VOCs was analyzed by GC-MS (Agilent 7890A GC coupled with a 5975C mass spectrometer (MS)). Twisters

were placed in the TDU at an initial temperature of 30 °C, which was enhanced at a rate of 30 °C/min to 210 °C and held for 10 min. The $N_2$-cooled injection system had an initial temperature of −50 °C during desorption. Splitless injection of the apple blossom bud volatiles into the GC was performed by heating at a rate of 12 °C/s to 220 °C that was held for 5 min. Chemical separation in the GC was achieved using a DB-5 ms column, 30 m × 250 µm × 0.25 µm (Agilent Technologies, Santa Clara, CA). The oven was programmed from 40 °C (held for 5 min) to 260 °C at 5 °C/min. A transfer line set at 280 °C led to the MS. The MS was operated in electron impact mode (70 eV). Helium was used as carrier gas (constant flow 1 ml/min). The mass range was 35 to 400 m/z at a scan time of 1.0 s. The MS source was set to 230 °C and the MS quad set to 150 °C. A standard mix of C8–C20 alkanes was analyzed using the same GC–MS method to calculate the Kovats Retention Indices of apple blossom bud volatiles. These were compared to known values as reported on the NIST website (*NIST Chemistry WebBook, 2002*). The identification of the detected compounds was based on their relative retention times and their mass spectra in comparison with those observed for pure standard substances. The other compounds were tentatively identified by comparison of mass spectra and retention indices (calculated according to *Van den Dool & Kratz (1963)*) with those recorded in the Adams and NIST mass spectral databases and the previously published data (*NIST Chemistry WebBook, 2002*; *Adams, 2014*). Compounds for which proper identification was not possible were defined as "unknown" and included in the analysis. The relative proportions of VOCs were calculated by setting the sum of all selected compounds to 100%. Only compounds unique to plant samples and not found in controls, *i.e.,* twisters in empty enclosures, and that had signal-to-noise ratios higher than 10 were retained.

## Data analysis

Proportion data including infestation rate and sex ratio of *A. pomorum* were modelled using generalized linear models (GLM) with binomial errors by analysis of deviance (categorical explanatory variable: effects of tree species). A quasibinomial error distribution was used if data were overdispersed with residual deviance >1.2 times greater than residual degrees of freedom (*Crawley, 2012*).

The influence of tree species and weevil sex on weevil body mass was analyzed by non-parametric Kruskal-Wallis test as normal distribution and homoscedasticity of the data could not be confirmed.

Differences in weevil body mass between males and females were analyzed by non-parametric Mann–Whitney $U$ test as normal distribution of the data could not be confirmed.

The choice of weevils in bioassays was analyzed with one-sample $chi^2$ tests, with counts of individuals choosing apple twigs with blossom buds or the control respectively.

Differences in proportions of individual compounds among the different apple species obtained by GC-MS analysis were compared by Mann–Whitney $U$ tests corrected by Benjamini–Hochberg. All data analyses were conducted with R, version 3.3.1 (*R Core Team, 2016*). Additionally, the chemical dataset was analyzed by a principal component analysis (PCA) (*Wold, Esbensen & Geladi, 1987*) using the software SIMCA-P, version

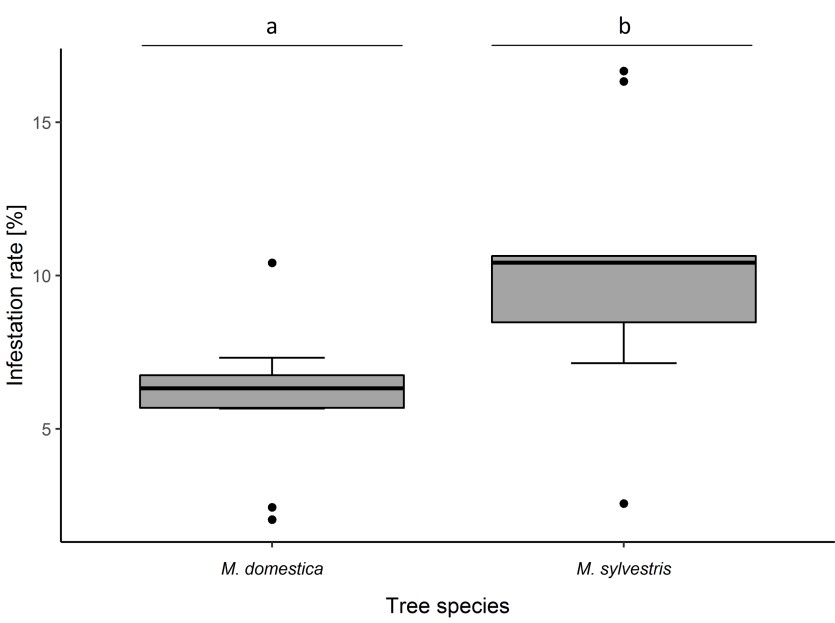

**Figure 1** **Infestation rates (%) of blossom buds of *Malus domestica* and *M. sylvestris*.** Different lower-case letters (a, b) indicate statistically significant differences (analysis of deviance; $F_{1,18} = 6.02$, $p = 0.016$). Number of sampled trees: *M. domestica*: 10, *M. sylvestris*: 10. Number of infested/sampled buds: *M. domestica*: 28/469 (5.97 ± 2.26% (mean ± SD) infested), *M. sylvestris*: 49/475 (10.32 ± 3.91% infested).

14.1 (Umetrics AB, Umeå, Sweden). The PCA was used to compare the volatile patterns of the different *M. sylvestris* genotypes and *M. domestica* cultivars with respect to the relative quantities of their volatile compounds (relative to the sum of quantities of all compounds). Data were log-transformed, mean-centered, and scaled to unit variance before being subjected to the analysis. The results of a PCA are usually discussed in terms of the loading plot, which describes the relationships between the variables with regard to the PCs (*Eriksson et al., 2001*).

## RESULTS

### Infestation rate by *A. pomorum*

Infestation rate of blossom buds was significantly higher in *M. sylvestris* than in *M. domestica* ($F_{1,18} = 6.02$, $p = 0.016$). Likewise, percentage infestation was higher in *M. sylvestris* (10.32 ± 3.91% (SD) infested blossom buds per tree) than in *M. domestica* (5.97 ± 2.26%) (Fig. 1).

Infestation rate also varied remarkably between cultivars of *M. domestica* and genotypes of *M. sylvestris* (Fig. S2) but was not tested statistically as only one individual per cultivar/genotype was assessed (47.2 ± 3.94 (mean ± SD) blossom buds per tree were examined). Percentage infestation was lowest in the *M. domestica* cultivar, Blauacher Wädenswil (2.1% infested blossom buds), followed by Kaiser Wilhelm (2.5%). The highest percentage infestation was recorded in two *M. sylvestris* genotypes: M. syl. 4 (16.7%)

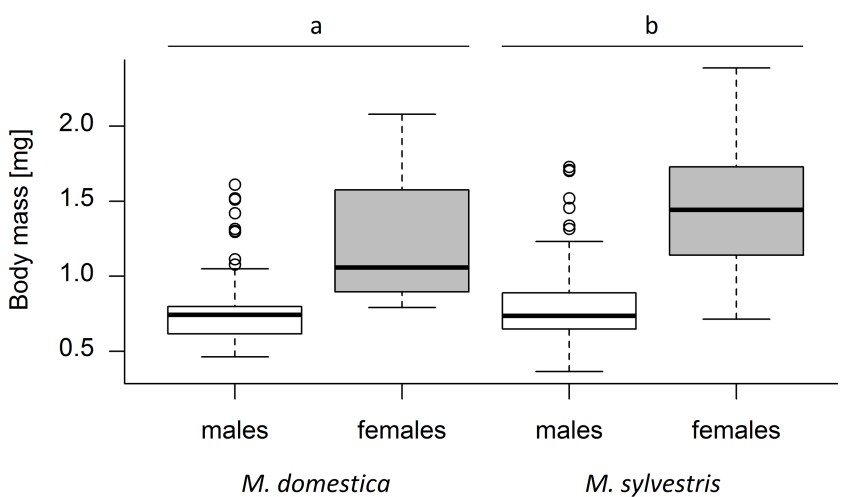

**Figure 2** **Dry body mass (mg) of male and female *Anthonomus pomorum* emerging from *Malus domestica* and *M. sylvestris*.** Different lowercase letters (a, b) indicate statistically significant differences (Kruskal-Wallis test; $\chi^2 = 10.30$, $p = 0.0013$). Number of sampled weevils: *M. domestica*: 139 (males: 74, females: 65), *M. sylvestris*: 281 (males: 132, females: 149).

that grew inside the botanical garden, and Eimersmühle 2 (16.3%) that grew outside the botanical garden in a riverbank (Fig. S2).

## Weevil body mass and sex ratio

A total of 420 weevils emerged from the capped blossoms (*M. domestica*: 139, *M. sylvestris*: 281). An average of $22.10 \pm 14.06$ (mean $\pm$ SD) weevils emerged from each study tree. Tree species and weevil sex both had a significant effect on weevil body mass (Kruskal-Wallis test; tree species: $\chi^2 = 10.30$, $p = 0.0013$; weevil sex: $\chi^2 = 220.81$, $p < 0.001$). Weevil body mass differed significantly between male and female weevils (Mann–Whitney $U$ test; $U = 75055$, $r = 0.40$, $p < 0.001$), with females being on average 69.6% heavier than males. The average body mass of weevils emerging from *M. sylvestris* (mean $\pm$ SD; males: $0.80 \pm 0.24$ mg, females: $1.45 \pm 0.36$ mg) was 15.2% higher than the mass of those emerging from *M. domestica* (males: $0.78 \pm 0.25$ mg, females: $1.23 \pm 0.40$ mg) (Fig. 2).

Tree cultivar/genotype also had a significant effect on weevil mass ($F_{1,18} = 13.39$, $p < 0.001$). The heaviest weevils were those emerging from *M. sylvestris* genotype Lochau 4 (males: $1.12 \pm 0.44$ mg, females: $1.95 \pm 0.22$ mg) and *M. domestica* cultivar James Grieve (males: $1.45 \pm 0.16$ mg, females: $1.86 \pm 0.17$ mg). The lightest weevils emerged from *M. domestica* cultivars Kaiser Wilhelm (males: 0.60 mg, females: 0.82 mg) and Hauxapfel (males: $0.64 \pm 0.10$ mg, females: $0.88 \pm 0.07$ mg) (Fig. S3).

Sex ratio was balanced. The percentage of females was higher in *M. sylvestris* (53% of 281 weevils) than in *M. domestica* (47% of 139 weevils) but no significant effect of tree species on sex ratio was detected ($F_{1,17} = 1.46$, $p = 0.23$).

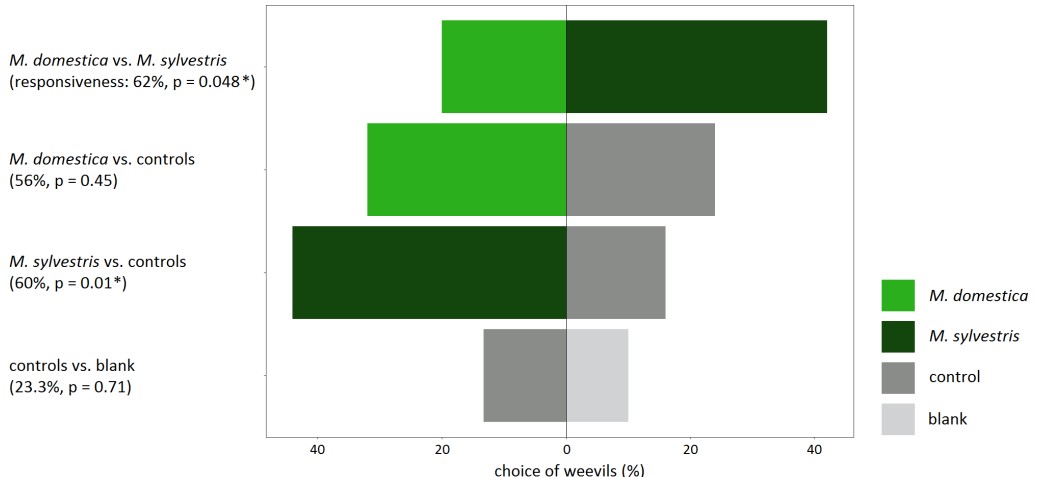

**Figure 3** **Response of field-collected *Anthonomus pomorum* in still-air, dual-choice olfactory bioassays to the odor of blossom buds of *Malus domestica* and *M. sylvestris*..** Both apple species were tested against each other, against controls as well as controls *vs.* blank. Sample sizes: $n = 50$, except for controls *vs.* blank: $n = 30$. $\chi^2$ test: An asterisk (*) indicates $p < 0.05$.

## Olfactory bioassays

Responsiveness, meaning the percentage of weevils that made a choice for either odor source, was high throughout all experiments that involved a treatment in the form of apple twigs with blossom buds (56–62%). When no blossom buds were tested and weevils could only choose between the wire control and a blank, the responsiveness was lowest (23.3%) (Fig. 3).

Weevils that were tested in bioassays for their preference for volatiles from apple blossom buds *versus* controls, significantly preferred *M. sylvestris* ($\chi^2 = 6.53$; $p = 0.01$), but not *M. domestica* over the controls ($\chi^2 = 0.57$; $p = 0.45$). Likewise, weevils that could choose directly between odors from *M. domestica* and *M. sylvestris*, significantly preferred *M. sylvestris* ($\chi^2 = 3.90$; $p = 0.048$). When the controls were tested against blanks to assure that they did neither have an attracting nor a repellent effect on weevils, responsiveness was lowest, and weevils did not prefer one over the other ($\chi^2 = 0.14$; $p = 0.71$) (Fig. 3).

## Analysis of headspace VOCs

Most compounds were emitted in higher relative amounts from blossom buds of *M. domestica* cultivars compared to *M. sylvestris* genotypes (Table 2). Acetophenone was the most emitted compound from blossom buds of *M. domestica* (25.5%) and *M. sylvestris* (24.0%), followed by pyridine (*M. domestica*: 12.2%, *M. sylvestris*: 10.7%). There were significant differences for phenylacetonitrile (emitted only from *M. domestica*), germacrene D and (-)-bornyl acetate (and five unidentified compounds) and tentative differences for $\alpha$-cadinene and $\gamma$-elemene. Linalool, cis-3-hexenyl isovalerate, and benzaldehyde were tentatively emitted in higher amounts from blossom buds of *M. sylvestris* (Table 2).

When comparing the blossom bud volatiles of *M. domestica* cultivars Jonagold and Topaz to those of *M. sylvestris* genotype Eimersmühle 2 that showed highly different

**Table 2 Variation of the phytochemical compositions (%) of blossom bud headspace VOCs among *Malus sylvestris* and *M. domestica*.**

| N | Components | GR[a] | RI[b] | *Malus sylvestris* | | *Malus domestica* | | P |
|---|---|---|---|---|---|---|---|---|
| | | | | Mean | SD | Mean | SD | |
| 1 | Pyridine | AR | <800 | 9.02 | 10.82 | 12.85 | 12.24 | n.s. (0.09) |
| 2 | Nonane | A | 900 | 2.81 | 5.05 | 3.75 | 4.79 | n.s. (0.08) |
| 3 | α-Pinene | MT | 931 | 1.04 | 2.71 | 1.43 | 2.44 | n.s. |
| 4 | β-Myrcene | MT | 990 | 0.73 | 1.29 | 0.98 | 1.22 | n.s. |
| 5 | Hexanoic acid | FA | 992 | 0.40 | 0.82 | 0.27 | 0.51 | n.s. |
| 6 | U1 | | 993 | 0.07 | 0.22 | 0.20 | 0.33 | n.s. |
| 7 | U2 | | 999 | 0.87 | 2.39 | 0.16 | 0.31 | n.s. |
| 8 | α-Phellandrene | MT | 1000 | 0.07 | 0.22 | 0.41 | 0.91 | n.s. |
| 9 | (Z)-3-Hexen-1-yl acetate | E | 1005 | 2.57 | 2.98 | 1.82 | 2.66 | n.s. |
| 10 | p-Cymene | ARMT | 1021 | 0.20 | 0.56 | 0.34 | 0.56 | n.s. |
| 11 | D-Limonene | MT | 1024 | 0.28 | 0.78 | 0.39 | 0.67 | n.s. |
| 12 | 2-Ethyl-1-hexanol | OH | 1028 | 0.48 | 0.61 | 0.54 | 0.48 | n.s. |
| 13 | Propyl tiglate | E | 1034 | 1.14 | 1.44 | 1.42 | 1.42 | n.s. |
| 14 | (E)- β-Ocimene | MT | 1047 | 0.84 | 1.21 | 1.87 | 3.53 | n.s. |
| 15 | γ-Terpinene | MT | 1056 | 0.11 | 0.41 | 0.17 | 0.34 | n.s. |
| 16 | Acetophenone | AR | 1061 | 18.11 | 7.92 | 19.19 | 7.10 | * |
| 17 | Linalool | OMT | 1099 | 5.64 | 5.68 | 2.61 | 1.87 | n.s. |
| 18 | Nonanal | Al | 1100 | 1.25 | 1.06 | 1.01 | 0.58 | n.s. |
| 19 | 2-Phenylethanol | AR | 1110 | 0.20 | 0.38 | 0.33 | 0.44 | n.s. |
| 20 | (E)-4,8-Dimethylnona-1,3,7-triene | HT | 1116 | 0.52 | 0.96 | 0.61 | 1.49 | n.s. |
| 21 | Phenylacetonitrile | AR | 1134 | 0.00 | 0.00 | 0.18 | 0.49 | n.s. |
| 22 | 3,3-Dimethylheptanoic acid | FA | 1148 | 0.33 | 0.53 | 0.72 | 1.19 | n.s. |
| 23 | p-Cymen-8-ol | MT | 1182 | 0.26 | 0.34 | 0.23 | 0.40 | n.s. |
| 24 | Methyl Salicylate | AR | 1190 | 0.46 | 0.57 | 0.52 | 0.71 | n.s. |
| 25 | Hexyl butanoate | E | 1192 | 0.14 | 0.24 | 0.27 | 0.35 | n.s. |
| 26 | Ethyl octanoate | E | 1197 | 0.05 | 0.14 | 0.11 | 0.18 | n.s. |
| 27 | Dodecane | Al | 1200 | 0.32 | 0.51 | 0.25 | 0.33 | n.s. |
| 28 | Decanal | Al | 1204 | 2.27 | 2.39 | 1.65 | 1.36 | n.s. |
| 29 | β-Phenoxyethanol | OH | 1218 | 2.47 | 2.92 | 1.63 | 1.31 | n.s. |
| 30 | cis-3-Hexenyl isovalerate | E | 1232 | 0.08 | 0.22 | 0.04 | 0.14 | n.s. |
| 31 | U3 | | 1271 | 0.50 | 0.29 | 0.37 | 0.28 | n.s. |
| 32 | U4 | | 1276 | 0.09 | 0.20 | 0.02 | 0.10 | n.s. |
| 33 | U5 | | 1280 | 0.31 | 0.50 | 0.60 | 0.79 | * |
| 34 | Bicyclo[2.2.1]heptan-2-ol, 1,7,7-trimethyl-, acetate, (1S-endo)- | E | 1283 | 0.08 | 0.25 | 0.02 | 0.10 | n.s. |
| 35 | Tridecane | A | 1300 | 2.32 | 2.73 | 2.75 | 3.39 | n.s. |
| 36 | U7 | | 1302 | 0.12 | 0.26 | 0.09 | 0.32 | n.s. |
| 37 | U8 | | 1305 | 0.29 | 0.71 | 0.32 | 0.65 | n.s. |
| 38 | U9 | | 1313 | 0.09 | 0.28 | 0.18 | 0.55 | n.s. |
| 39 | U10 | | 1319 | 0.13 | 0.42 | 0.22 | 0.47 | n.s. |

**Table 2** (*continued*)

| N | Components | GR[a] | RI[b] | *Malus sylvestris* | | *Malus domestica* | | P |
|---|---|---|---|---|---|---|---|---|
| | | | | **Mean** | **SD** | **Mean** | **SD** | |
| 40 | U11 | | 1323 | 0.08 | 0.20 | 0.14 | 0.37 | n.s. |
| 41 | U12 | | 1327 | 0.88 | 0.84 | 1.10 | 1.55 | n.s. |
| 42 | $\gamma$-Elemene | ST | 1335 | 0.51 | 1.15 | 0.58 | 1.40 | n.s. |
| 43 | U13 | | 1342 | 0.02 | 0.09 | 0.09 | 0.17 | n.s. |
| 44 | U14 | | 1343 | 0.39 | 0.95 | 0.49 | 1.07 | n.s. |
| 45 | U15 | | 1346 | 0.28 | 0.91 | 0.38 | 0.98 | n.s. |
| 46 | U16 | | 1354 | 0.12 | 0.25 | 0.24 | 0.58 | n.s. |
| 47 | U17 | | 1361 | 6.02 | 10.40 | 7.20 | 10.93 | * |
| 48 | U18 | | 1364 | 0.32 | 1.10 | 0.25 | 0.65 | n.s. |
| 49 | 3-Methyl-tridecane | A | 1370 | 0.11 | 0.25 | 0.07 | 0.22 | n.s. |
| 50 | ß-Bourbonene | ST | 1384 | 0.77 | 1.75 | 0.40 | 0.68 | n.s. |
| 51 | U19 | | 1387 | 1.65 | 2.75 | 1.69 | 2.17 | * |
| 52 | U20 | | 1394 | 1.06 | 1.94 | 0.95 | 1.32 | n.s. |
| 53 | Dodecanal | Al | 1407 | 3.72 | 4.13 | 1.46 | 1.28 | n.s. |
| 54 | U21 | | 1426 | 0.05 | 0.19 | 0.30 | 0.91 | n.s. |
| 55 | ß-Copaene | ST | 1446 | 0.93 | 2.27 | 0.33 | 0.52 | n.s. |
| 56 | U22 | | 1448 | 0.90 | 0.96 | 0.69 | 0.81 | n.s. |
| 57 | Geranyl Acetone | K | 1450 | 0.88 | 1.78 | 0.60 | 1.11 | n.s. |
| 58 | U23 | | 1451 | 0.43 | 0.90 | 0.86 | 1.45 | n.s. |
| 59 | p-Benzoquinone, 2,6-di-tert-butyl- | MMT | 1464 | 0.61 | 0.54 | 0.38 | 0.33 | n.s. |
| 60 | U24 | | 1474 | 1.79 | 6.50 | 0.64 | 1.65 | n.s. |
| 61 | U25 | | 1474 | 0.45 | 1.20 | 0.83 | 1.67 | n.s. |
| 62 | $\gamma$-Muurolene | ST | 1475 | 0.74 | 2.89 | 0.17 | 0.25 | n.s. |
| 63 | U26 | | 1478 | 0.47 | 0.52 | 0.37 | 0.56 | n.s. |
| 64 | Germacrene D | ST | 1479 | 2.73 | 4.87 | 1.91 | 2.36 | n.s. |
| 65 | 1-Pentadecene | AE | 1491 | 0.59 | 1.60 | 0.47 | 0.86 | n.s. |
| 66 | Bicyclogermacrene | ST | 1495 | 0.36 | 0.80 | 0.49 | 1.45 | n.s. |
| 67 | U27 | | 1495 | 1.40 | 3.06 | 1.30 | 2.10 | n.s. |
| 68 | Pentadecane | A | 1500 | 0.94 | 1.24 | 0.87 | 0.73 | n.s. |
| 69 | $\alpha$-Farnesene | ST | 1508 | 6.16 | 7.05 | 6.39 | 11.54 | n.s. |
| 70 | $\gamma$-Cadinene | ST | 1512 | 0.19 | 0.38 | 0.21 | 0.34 | n.s. |
| 71 | $\delta$-Cadinene | ST | 1523 | 0.27 | 0.44 | 0.30 | 0.35 | n.s. |
| 72 | $\alpha$-Cadinene | ST | 1536 | 0.85 | 1.54 | 1.09 | 2.08 | n.s. |
| 73 | U28 | | 1541 | 1.69 | 3.81 | 1.71 | 2.74 | * |
| 74 | U29 | | 1549 | 0.41 | 1.40 | 0.43 | 0.91 | n.s. |
| 75 | U30 | K | 1552 | 0.55 | 1.78 | 0.63 | 1.15 | * |

**Notes.**
[a]GR = group of chemical compounds (A, alkane; AR, aromatic compound; Al, aldehyde; ARMT, aromatic monoterpene; E, ester; FA, fatty acid conjugate; K, ketone; MMT, monocyclic monoterpenoid; MT, monoterpene; OH, alcohol; OMT, oxygenated monoterpene; ST, sesquiterpenoid; HT, homoterpene).
[b]RI = retention index (DB5—fused silica capillary column 30 m × 0.25 mm i.d., 0.25 μm film thickness) experimentally determined using a homologue series of n-alkanes.
[a]GR = group of chemical compounds (A, alkane; AE, alkene; K, ketone; MMT, monocyclic monoterpenoid; ST, sesquiterpenoid).
[b]RI = retention index (DB5—fused silica capillary column 30 m × 0.25 mm i.d., 0.25 μm film thickness) experimentally determined using a homologue series of n-alkanes.

infestation rates in the field (see Fig. S2), linalool was emitted at 4–5-fold higher rates from Eimersmühle 2 blossom buds (10.8%) compared to Jonagold and Topaz (2.4%) (Table S3). Also, 1-pentadecene and $\gamma$-cadinene were emitted from *M. sylvestris* genotype Eimersmühle 2 at higher rates (1.5%) compared to other genotypes/cultivars. Comparisons between VOCs emitted by blossom buds of similarly infested *M. domestica* cultivars Jonagold/Topaz and *M. sylvestris* genotype Lochau 1 did not reveal significant differences in their volatile bouquet (Table S3). However, similarly low infested *M. domestica* cultivars Jonagold/Topaz and *M. sylvestris* genotype Lochau 1 all emitted higher relative amounts of acetophenone, pyridine, and nonane compared to the more infested "Eimersmühle 2" (Table S3).

The PCA (Fig. 4) explained only a total of 28.2% of the variance of the data with 17.0% by the first and 11.2% by the second principal component. The score plot (Fig. 4A) of the relative amount of volatile compounds emitted by the two different apple species shows that the *M. domestica* cultivars and the *M. sylvestris* genotypes cannot be separated with respect to the relative amount of volatile compounds in total. While the *M. domestica* cultivar Golden Delicious or the *M. sylvestris* genotype Lochau 1 build clusters, others do not cluster together based on their identified blossom bud volatiles. In the loading plot, variables not explaining the separation of the clusters are generally located toward the zero origin and the more important variables are located at the periphery of the plot (Fig. 4B). The locations of sesquiterpenes like $\alpha$-cadinene, $\delta$-cadinene, and germacrene D on the top of PC2 indicate important variables that are separating Golden Delicious plants and other *M. domestica* cultivars from the cluster of *M. sylvestris* genotypes, which are characterized by linalool emission.

## DISCUSSION

A few studies have been published so far that examine the infestation rate and resistance of different cultivars of *M. domestica* to the apple blossom weevil *A. pomorum* (*Kalinová et al., 2000*; *Mody, Spoerndli & Dorn, 2011*; *Mody, Collatz & Dorn, 2015*). However, to our knowledge, only one study has previously addressed the susceptibility and suitability of different *Malus* species, including the European crab apple *M. sylvestris*, to *A. pomorum* (*Knuff, Obermaier & Mody, 2017*). In terms of olfactory host plant choice of *A. pomorum*, *M. sylvestris* as the only apple species native to central Europe has not been investigated at all.

The results of our comparative study confirm the findings of *Knuff, Obermaier & Mody (2017)* that female *A. pomorum* show specific host selection patterns among *Malus* species, which is indicated by significant differences in infestation rates between *M. domestica* and *M. sylvestris*. The European crab apple showed significantly higher infestation by *A. pomorum* than the domesticated apple. Differences in weevil mass confirm the previous findings that the two *Malus* species vary in their suitability as hosts (*Knuff, Obermaier & Mody, 2017*). However, in contrast to the results of *Knuff, Obermaier & Mody (2017)*, weevils emerging from *M. domestica* were significantly lighter than those originating from *M. sylvestris*. These results imply that *M. domestica* blossom buds offer decreased suitability

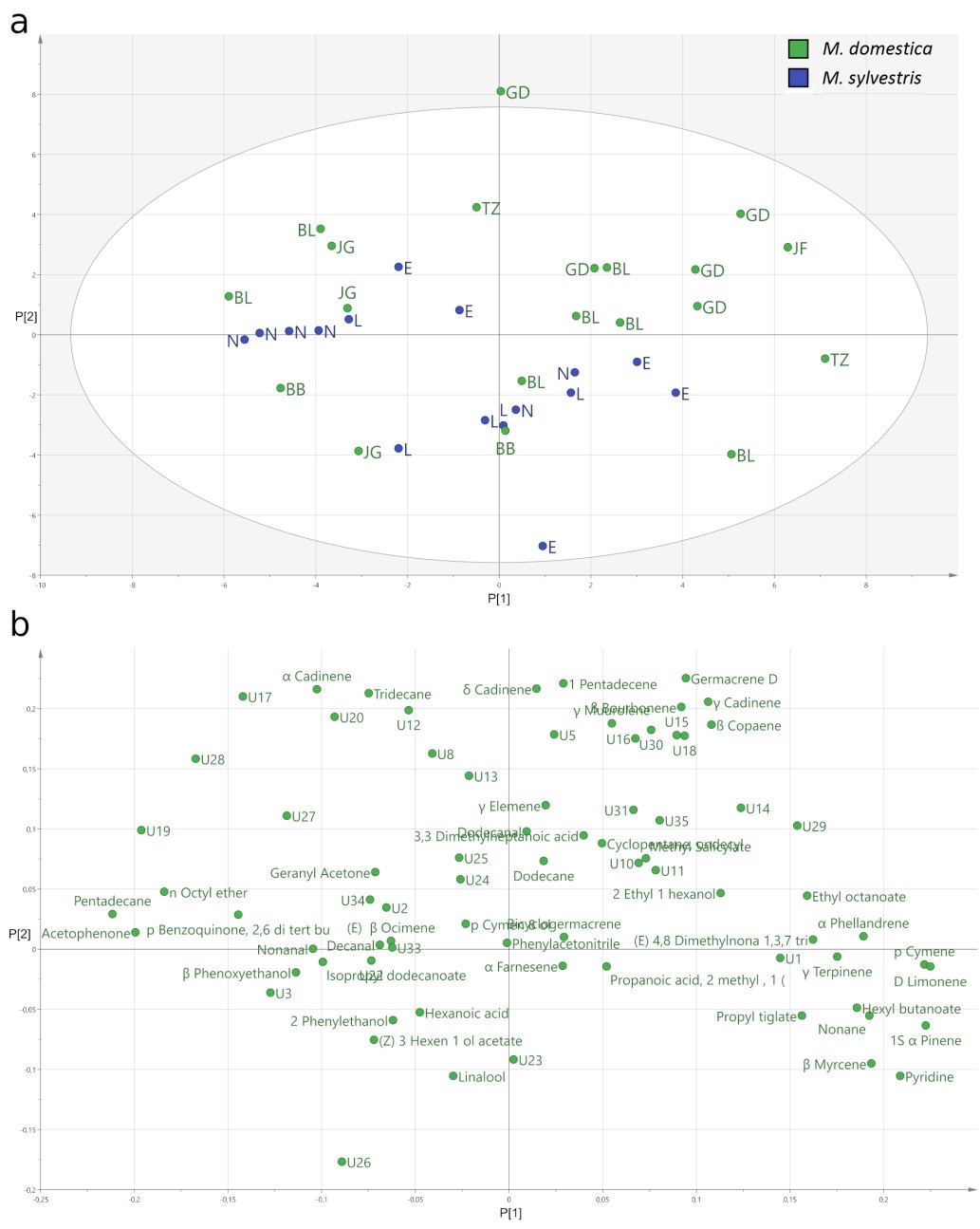

**Figure 4** **Principal components analysis (PCA) of the headspace VOCs emitted from blossom buds of** ***Malus domestica*** **and** ***M. sylvestris.*** (A) Score plot of the relative amounts of VOCs emitted from blossom buds of *M. domestica* (green) and *M. sylvestris* (blue). (B) Loading plot of the relative amount of VOCs emitted from blossom buds of *M. domestica* and *M. sylvestris*.

for *A. pomorum*, meaning a higher antibiosis resistance (*Mody, Collatz & Dorn, 2015*) compared to *M. sylvestris*. In addition, trees of *M. domestica* also proved less attractive to *A. pomorum*, meaning a higher antixenosis resistance (*Mody, Collatz & Dorn, 2015*), as indicated by both less attractive odors of blossom buds in bioassays and a lower infestation

rate compared to *M. sylvestris*. The analysis of headspace VOCs revealed differences in the blossom bud odors separating a group of *M. sylvestris* genotypes from *M. domestica* and another group of *M. sylvestris* genotypes which might be employed in further studies to repel *A. pomorum* from *M. domestica*.

## Infestation rate

As stated above, there were significant differences in infestation rate by *A. pomorum* between *M. domestica* and *M. sylvestris*. Showing that crab apple trees were significantly more infested than *M. domestica* trees, our study provides new evidence in support of earlier observations by *Knuff, Obermaier & Mody (2017)*. Infestation rate also varied remarkably between cultivars of *M. domestica*, as was already reported by other studies (*Kalinová et al., 2000*; *Mody, Spoerndli & Dorn, 2011*; *Mody, Collatz & Dorn, 2015*).

Overall percentage infestation of *M. domestica* was 6.0%, indicating that weevil populations in the Ecological-Botanical Garden of the University of Bayreuth were below the economic threshold level set at 10–15%, above which pest control is advised to prevent yield losses (*Höhn & Stäubli, 1989*). Trees of *M. sylvestris* showed an overall percentage infestation of 10.2%. Although economic threshold levels do not apply for wild apple species, it is nevertheless interesting to notice that weevil populations on *M. sylvestris* trees reached the economic threshold level set for *M. domestica*.

The high infestation rate of *M. sylvestris* is remarkable, since wild apple individuals did not grow on a meadow surrounded by other apple trees like the *M. domestica* individuals used in this study. Except for the two individuals inside the botanical garden, *M. sylvestris* individuals grew in hedgerows, riverbanks and forest margins in the surroundings of Bayreuth where host density was much smaller compared to the apple orchard in the botanical garden. However, there are no indications that *A. pomorum* populations were generally lower inside the botanical garden than outside, as one of the *M. sylvestris* individuals inside the botanical garden was the most infested tree in this study (M. syl. 4: 16.7% infested blossom buds).

What also makes the high infestation rate of *M. sylvestris* remarkable is the notion that domesticated crop plants are generally assumed to be more heavily infested by pest insects than their wild relatives due to tradeoffs between growth and defense against herbivores (*Strauss et al., 2002*; *Stoeckli et al., 2009*; *Kempel et al., 2011*; *Stoeckli et al., 2011*; *Huot et al., 2014*). Implementation of effective defense mechanisms imposes a substantial demand for resources, which has been suggested to reduce growth (*Huot et al., 2014*). However, domesticated crop plants like *M. domestica* are bred for high yield and fruits that are larger and sweeter than those of their wild relatives, which requires an increased resource allocation to growth processes (*Strauss et al., 2002*; *Kaplan, Dively & Denno, 2009*; *Stoeckli et al., 2011*). This tradeoff mechanism usually reduces antiherbivore defense in domesticated crop plants making them more susceptible to herbivores (*Koricheva, 2002*; *Strauss et al., 2002*; *Kaplan, Dively & Denno, 2009*; *Kempel et al., 2011*). This mechanism was not observed in our study, as illustrated by the higher infestation rate of *M. sylvestris* trees compared to *M. domestica*. But what could explain the higher susceptibility of *M. sylvestris* trees to *A. pomorum*?

Distance to hedgerows is apparently not a significant factor for the lower infestation rate among *M. domestica* trees as the two *M. domestica* trees closest to the hedgerow in the south, namely the individuals Gala and Jonathan type 'Watson' were not among the most infested *M. domestica* trees. Temperature and site-dependent effects could mostly be ruled out since *M. sylvestris* trees inside and outside of the botanical garden were both on average more infested than the *M. domestica* trees inside the botanical garden. Therefore, it seems that *M. sylvestris* trees possess certain traits that attract *A. pomorum* and guide females in search of their host plant. Since *A. pomorum* is attracted by blossom buds, the traits of these buds have previously been pointed out to be particularly promising for explaining differences in susceptibility to *A. pomorum* (*Mody, Collatz & Dorn, 2015*).

The finding that tree species affected infestation rates indicates that there is at least one species-dependent variable that has not been investigated (*Knuff, Obermaier & Mody, 2017*). This variable could be the composition of headspace VOCs emitted by the blossom buds. VOCs have been described as a promising trait mediating the preference of host-searching female weevils (*Mody, Collatz & Dorn, 2015*), as they may provide leading cues for females in search of a suitable oviposition site (*Piskorski & Dorn, 2010*; *Collatz & Dorn, 2013*; *Mody, Collatz & Dorn, 2015*).

### Weevil body mass and sex ratio

Weevils emerging from *M. sylvestris* had a significantly higher body mass than those originating from *M. domestica*, indicating a higher suitability of *M. sylvestris* for the weevil offspring. According to the preference-performance hypothesis, this finding fits quite well to the higher infestation rate of *M. sylvestris* observed in this study. *Knuff, Obermaier & Mody (2017)*, in contrast, did not detect significant differences in body mass of weevil offspring originating from *M. sylvestris* or *M. domestica*. In our study, significant differences in weevil mass were also identified between male and female weevils for both tree species. Female weevils emerging from the blossom buds had a higher body mass than males. This female-biased sexual size dimorphism (SSD) was consistent with results of previous studies (*Mody, Collatz & Dorn, 2015*; *Knuff, Obermaier & Mody, 2017*). We also detected significant differences in weevil mass between single cultivars of *M. domestica* or genotypes of *M. sylvestris*, which is consistent with the findings of *Mody, Collatz & Dorn (2015)* who reported significant differences in body mass of weevils that originated from different cultivars of *M. domestica*. This confirms the notion that cultivars of *M. domestica* that differ in their nutritional composition also differ in their suitability for *A. pomorum* (*Mody, Collatz & Dorn, 2015*). The same might be true for different genotypes of *M. sylvestris*.

Preference-performance relationships are regularly expected, although they are not necessarily detected in nature (*Mody, Collatz & Dorn, 2015*), probably due to imperfect adaptations, imperfect decisions and a prolonged decision time especially in polyphagous herbivore species (*Bernays, 2001*; *Gripenberg et al., 2010*; *Mody, Collatz & Dorn, 2015*). This study showed that *M. sylvestris* that is native to Central Europe and offers a higher nitrogen content in its blossom buds compared to *M. domestica* (*Knuff, Obermaier & Mody, 2017*), was significantly more infested by the apple blossom weevil, indicating a higher preference of *A. pomorum* for this species. Furthermore, weevil offspring developing within

*M. sylvestris* blossom buds had a significantly higher body mass than weevils developing within *M. domestica* blossom buds, indicating also a better performance of *A. pomorum* larvae on *M. sylvestris*. Hence, this study confirms the findings of *Mody, Collatz & Dorn (2015)* that preference-performance relationships may play a role in infestation of *Malus* cultivars by the oligophagous weevil *A. pomorum* and indicates that this is also true across different *Malus* species.

## Olfactory orientation

This study represents the first examination of the behavioral response of apple blossom weevils to the headspace VOCs of *M. sylvestris* blossom buds. It has been shown in previous studies that apple blossom weevils were attracted to the odor of *M. domestica* blossom buds, and therefore most likely rely on olfactory cues to locate their host plant (*Piskorski & Dorn, 2010*; *Collatz & Dorn, 2013*). However, behavioral response to the odor of blossom buds of wild apple species has not been tested yet. Composition of headspace VOCs emitted by the blossom buds might explain the higher infestation rate by *A. pomorum* of *M. sylvestris* in comparison to *M. domestica* that was observed in this study.

The olfactory response of field-collected apple blossom weevils to headspace VOCs of both *Malus* species showed that VOCs emitted from blossom buds of *M. sylvestris* were significantly more attractive to *A. pomorum* than VOCs emitted from *M. domestica* blossom buds. This finding was obtained when both *Malus* species were tested against each other in the same setup, but also when blossom buds of each apple species were tested against the wire controls. When controls were tested against blanks, results showed that controls had neither an attractant nor a repellent effect on weevils. Responsiveness, meaning the percentage of weevils making a choice in the dual-choice bioassays, was sufficiently high throughout the experiments, and at a similar level to that reported for other herbivores responding to natural samples or synthetic blends of host plant-derived odors (*Prokopy, Cooley & Phelan, 1995*; *Piñero & Dorn, 2007*; *Collatz & Dorn, 2013*).

VOCs emitted from prebloom *M. domestica* blossom buds that attracted *A. pomorum* have been identified (*Piskorski & Dorn, 2010*), and bioassays with nature-identical synthetic compounds have been successfully conducted (*Collatz & Dorn, 2013*). The finding of the present study, that VOCs emitted from *M. sylvestris* blossom buds were significantly more attractive than those emitted from *M. domestica*, might favor the successful use of *M. sylvestris*-derived synthetic volatiles in monitoring and management of *A. pomorum*. It might also offer the opportunity to use *M. sylvestris* as trap crop in apple plantations when establishing push-pull systems.

## Headspace VOCs

Prior to the testing of *M. sylvestris*- or *M. domestica*-derived synthetic volatiles for monitoring or management of *A. pomorum* populations in commercial apple orchards, VOCs have to be identified that are responsible for the increased attractive effect of *M. sylvestris* blossom buds or the decreased attractivity of *M. domestica* ones. Although a complex blend of volatiles comprised of 16 VOCs from *M. domestica* blossom buds emitted in a suitable stage for oviposition by *A. pomorum* has already been described (*Piskorski &*

*Dorn, 2010*), knowledge on VOCs from apple blossom buds released before bloom is still scarce. Therefore, the chemical spectrum of VOCs from *M. sylvestris* blossom buds released before bloom has been analyzed here for the first time. We are aware that the results of the VOC analysis cannot readily be employed to explain the results from the field trials as the potted trees were much younger than the study trees in the botanical garden and in field sites which could influence their VOC profile. The fact that some cultivars/genotypes were used in the sampling of VOCs that have not been used for the field trials might also weaken the connection between the results of the VOC analysis and the field results. However, the results do serve as an indication of which VOCs might be of interest and further study is required.

It has been shown for many curculionids that they regularly rely on olfactory cues for host detection, among them close relatives of the apple blossom weevil, *e.g.*, the strawberry blossom weevil *Anthonomus rubi* Herbst (*Bichão et al., 2005*), or the boll weevil *Anthonomus grandis* Boheman (*Minyard et al., 1969*; *Dickens, 1989*). As *A. grandis* can become a devastating pest of cotton, it is by far the best-studied species of the genus *Anthonomus* (*Collatz & Dorn, 2013*). *Minyard et al. (1969)* reported attractant effects of cotton-square VOCs in bioassays conducted with *A. grandis*, as well as attractant effects of several single compounds identified in the blend, including the ubiquitous plant volatile *β*-caryophyllene.

Furthermore, it has recently been shown for two other species of curculionids, the plum curculio *Conotrachelus nenuphar* Herbst and the vine weevil *Otiorhynchus sulcatus* Fabricius, that traps, which were moderately effective became more powerful in attracting the target weevils after the addition of host-plant-derived VOCs (*Leskey, Zhang & Herzog, 2005*; *Van Tol et al., 2012*). A similar effect was reported for *A. grandis* when components of host-plant odor were added to traps baited with aggregation pheromones (*Dickens, 1989*). However, as pheromones of *A. pomorum* are unknown so far (*Dorn & Piñero, 2009*), host plant-derived VOCs are of increased relevance for the development of monitoring or management tools for this herbivore. Olfactory attraction to odor traps baited with a synthetic blend of host plant-derived VOCs holds the potential to substantially increase the number of captured individuals and might lead to an improved monitoring system (*Collatz & Dorn, 2013*).

Since headspace VOCs have been shown to differ among cultivars of *M. domestica* (*Kalinová et al., 2000*; *Hern & Dorn, 2003*), they can be expected to differ even more among *Malus* species and may provide leading cues for females in search of a suitable oviposition site (*Piskorski & Dorn, 2010*; *Collatz & Dorn, 2013*; *Mody, Collatz & Dorn, 2015*).

The present study is the first that compares headspace VOCs of blossom buds of different *Malus* species before bloom and presents the first report of headspace VOCs of *M. sylvestris* sampled at early phenological stages. This enables a comparison between headspace VOCs of *M. sylvestris* and *M. domestica*. The collection of headspace VOCs from both apple species using stir bar sorptive extraction (SBSE) with Twisters, followed by thermal desorption and GC-MS analysis, allowed tentative identification of a total of 47 volatile compounds, while 30 compounds remained unidentified. Methodological differences likely account for a deviation from compounds identified by *Piskorski & Dorn (2010)* in *M. domestica*.
For example, in the present study we sampled more buds at later bud stages and at higher temperatures with a different technique and adsorbent.

The principal component analysis (PCA) revealed no clear separation of *M. domestica* cultivars and *M. sylvestris* genotypes based on blossom bud volatiles (Fig. 4). While plants from a few genotypes and cultivars cluster together with respect to their chemical profiles, the high variation of the composition of emitted compounds within other genotypes and cultivars conceals a clear pattern. Another reason for the variation might be slight differences in the phenology of the blossom buds. The chromatographic analysis of bud emanations from different phenological stages in the same cultivar revealed both quantitative and qualitative differences (*Kalinová et al., 2000*).

Acetophenone, the most-emitted compound of both *Malus* species, was also found in blossom buds or flowers of the wild Italian red apple (*Fraternale et al., 2014*), strawberries (*Mozūraitis et al., 2020*), or red clover (*Buttery, Kamm & Ling, 1984*). Acetophenone has a repellent effect on woodwasps, possibly preventing them from ovipositing in Mongolian pine infested by non-symbiotic fungi (*Wang et al., 2019*). Acetophenone can repel *Drosophila suzukii* from raspberries (*Renkema & Smith, 2020*) and can even have insecticidal effects (*Dettner et al., 1992*). Pyridine, the second most-emitted plant volatile from the blossom buds has rarely been detected from plant odors, but it was reported for grape leaves (*Giacomuzzi et al., 2017*), bearberry plants (*Radulović, Blagojević & Palić, 2010*), and floral volatiles of *Eriotheca longitubulosa* (*MacFarlane, Mori & Purzycki, 2003*) and is known to attract flies and thrips (*Łyczko et al., 2021*).

Most compounds were emitted in higher relative amounts from *M. domestica*. Phenylacetonitrile was emitted only from *M. domestica* and there were significant differences for germacrene D, (-)-bornyl acetate and tentative differences for $\alpha$-cadinene and $\gamma$-elemene. Phenylacetonitrile is known from apple tree volatiles induced by caterpillar feeding and attracting conspecific herbivores and a generalist predator (*El-Sayed et al., 2018*). The sesquiterpenes germacrene D, $\alpha$-cadinene, and $\gamma$-elemene emitted from *M. domestica* flower buds are known from emissions of different plants and can repel or kill insects at higher concentrations (*Benelli et al., 2018*; *Tholl, 2015*). Germacrene D and (-)-bornyl acetate are known from apple tree flower emissions and elicit responses in herbivores (*Bengtsson et al., 2001*; *Buchbauer et al., 1993*; *Yaqin & Shixiang, 2021*). Linalool was tentatively emitted in higher relative amounts from blossom buds of all genotypes of *M. sylvestris* and 4-5 times more from genotype Eimersmühle 2 (10.8%). It might have been even more detectable since the relative amounts of most other compounds were reduced. Linalool is known to be a main volatile (43%) of flower buds of red apples attracting pollinators (*Fraternale et al., 2014*). Linalool is also present in higher amounts in flower buds of cultivar Royal Gala apples, possibly providing protection of the reproductive organ against oxidative stress and pathogenic microbes (*Nieuwenhuizen et al., 2013*). According to *Nieuwenhuizen et al. (2013)* its production may thus be maintained under a positive selective pressure in wild apples, while the small number of terpenes found in modern cultivars may be related to commercial breeding strategies.

## CONCLUSIONS

This study confirmed the hypothesis that *A. pomorum*, an economically relevant pest insect in European apple orchards, shows specific host selection patterns among *Malus* species. Infestation rate of *A. pomorum* was significantly higher in the wild European crab apple *M. sylvestris*, indicating a preference of female weevils searching for a suitable host tree for oviposition. Furthermore, a higher average mass of weevils originating from *M. sylvestris* implies a higher suitability of the native *M. sylvestris* for the apple blossom weevil. These two findings combined strongly support the idea that preference-performance relationships might play a role for *A. pomorum*.

Olfactory bioassays conducted with blossom buds of both *Malus* species showed the weevils' preference for the complex odor of *M. sylvestris* blossom buds over VOCs emitted from blossom buds of *M. domestica*. This finding may explain the higher infestation rate of *M. sylvestris* trees in the field and may therefore represent the missing variable accounting for an increased susceptibility of *M. sylvestris* trees compared to *M. domestica*.

Therefore, an analysis of headspace VOCs emitted from blossom buds of both *Malus* species before bloom was conducted for *M. sylvestris* for the first time and candidate compounds for further study have been identified.

Olfactory attraction of apple blossom weevils in combination with other tools holds the potential to substantially increase the number of *A. pomorum* captured in shelter traps and might therefore constitute an improved monitoring system. Since pheromones are still unknown for this economically relevant pest insect (*Dorn & Piñero, 2009*), host plant-derived VOCs of *M. sylvestris* and *M. domestica* blossom buds represent promising cues in a multisensorial monitoring tool.

## ACKNOWLEDGEMENTS

We thank the team of gardeners in the Ecological-Botanical Garden, especially Guido Arneth, Claus Rupprich, Kerstin Günther, Annette Berthold, Helmut Zapf, and René Huber, for their help with tree propagation and maintenance. We also thank Gregor Aas and Martin Feulner for showing us verified *M. sylvestris* individuals at field sites around Bayreuth and Detlef Ulrich, René Grünwald, Ali Karimi (JKI), and Jona Höfflin (FU Berlin) for help with the GC-MS analyses of headspace VOCs.

### Funding

This work was funded by the Deutsche Forschungsgemeinschaft (DFG, German Research Foundation) (491183248) and by the Open Access Publishing Fund of the University of Bayreuth. The funders had no role in study design, data collection and analysis, decision to publish, or preparation of the manuscript.

### Grant Disclosures

The following grant information was disclosed by the authors:

Deutsche Forschungsgemeinschaft (DFG, German Research Foundation): 491183248. Open Access Publishing Fund of the University of Bayreuth.

## Competing Interests

The authors declare there are no competing interests.

## Author Contributions

- Benjamin Henneberg conceived and designed the experiments, performed the experiments, analyzed the data, prepared figures and/or tables, and approved the final draft.
- Torsten Meiners analyzed the data, prepared figures and/or tables, authored or reviewed drafts of the article, and approved the final draft.
- Karsten Mody conceived and designed the experiments, authored or reviewed drafts of the article, and approved the final draft.
- Elisabeth Obermaier conceived and designed the experiments, authored or reviewed drafts of the article, and approved the final draft.

## Data Availability

The data is available at Dryad Digital Repository: Henneberg, Benjamin; Meiners, Torsten; Mody, Karsten; Obermaier, Elisabeth (2022), Morphological and olfactory tree traits influence the susceptibility and suitability of the apple species *Malus domestica* and *M. sylvestris* to *Anthonomus pomorum*, Dryad, Dataset, https://doi.org/10.5061/dryad.v15dv41xr.

## Supplemental Information

Supplemental information for this article can be found online at http://dx.doi.org/10.7717/peerj.13566#supplemental-information.

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
