# Peer review of "Morphological and olfactory tree traits influence the susceptibility and suitability of the apple species Malus domestica and M. sylvestris to the florivorous weevil Anthonomus pomorum (Coleoptera: Curculionidae)"

_PeerJ, doi:10.7717/peerj.13566_

## Round 0.1 · original submission · Major Revisions

Dear Authors

I have received the comments from reviewers. On the behalf of their reviews article required major revisions, particularly, authors need to give attention to reviewer 1 comments. We are pleased to receive the revised version soon.

Reviewer 1 ·

Excellent Review

This review has been rated excellent by staff (in the top 15% of reviews)
EDITOR COMMENT
The evaluation of the article by the reviewer was excellent. In my career, a very few number of review reports have been outstanding and this may be considered as one of them. I believe that the reviewer's efforts will make the scientific material more worthwhile and reliable. I really enjoyed it.

Basic reporting

The authors have provided a very clear manuscript on an interesting area of research. There are no issues with the English. The literature review is extensive and mostly adequate. There are some references with issues (see below) and the introduction could benefit from more recent references, if applicable. Where this is not possible, authors should mention that the references provided are the most up to date. Structure of the article is fine. Manuscript is self-contained with results relevant to the hypotheses.

Bengtsson – not in text and reference not complete ‘….’ In the author list??
Cornille et al 2012 – ‘….’ In the author list for this reference
Dettner, 1992 – need ‘et al’ in the text and ‘…..’ in the author list?
Hausmann et al. 2005 – not in text
Lyczko et al. 2021 – has ‘….’ In author list
Meier et al. 2009 – has ‘….’ In author list
Mozuaritis et al. 2020 – has ‘…’ in author list
Toepfer et al. 2000 – not in text
Velasco et al. 2010 – has ‘….’ In author list

Experimental design

The research is primary in nature and fits within the research aims and scope of PeerJ. The research question is well defined and relevant to the tree fruit industry and definitely fills a knowledge gap. The investigation is multi-faceted involving both field and laboratory components. The field evaluations are reasonably well described. Where they fall down is that the authors fail to indicate why they chose the domesticated trees they did. The various cultivars are, I feel, an issue which the authors are trying to overlook, or assume that this is an irrelevant factor. If the authors could provide further information on the choice of trees it may help assauge the concern that their results are more than lucky. Were these trees chosen for their proximity to the only crab apple tree within the botanical garden? proximity to the hedgerows (and source of weevils)? Their choice of crab apple trees, being in locations other than the botanical garden, may play a role in their results as these could be experiencing a different population than the domesticated trees in the garden. For the lab experiments, the authors fail to mention which cultivars they used in the bioassay: a small set, many or all? Would be impossible to replicate this experiment with the information provided. The headspace collection (using potted trees) suffers from connectivity to the rest of the work, particularly how the authors have incorporated this data. Only 2 of the domesticated apple cultivars and 3 of the crabapple genotypes match with those used in the field trials. This makes the connection between the findings weak at best. I understand what the authors are trying to accomplish, but the material used provides, at best, an indication of which VOC are of interest and further study is required. A larger issue with the VOC work is that they captured volatiles from stages 56/57 (when oviposition has begun) yet the adults are selecting the trees 3 phenological stages earlier (at bud burst). It would be more biologically relevant to collect at this earlier stage when the adults are moving to the trees from the hedgerows, rather than at the stage where oviposition begins (or compare these profiles). The profile which indicates 'come hither' may differ from the profile which says 'commence oviposition'. The authors have captured the latter but not necessarily the former and this is a definite weakness in this aspect of the work.

Validity of the findings

All of the findings from the work suffer from the weaknesses described in the earlier section. If the authors can justify their selection of trees for the field study, and provide assurance that the trees were exposed to a roughly similar population pressure, then the results might stand. If there is no such assurance possible, e.g. the trees were not selected with consideration of distance from the hedgerows, then there are confounding factors (cultivar and distance) which could serve to explain their field results. For the laboratory trials, the lack of attention to the cultivars used and the failure to better connect the field results with the laboratory trials weakens these results as well.

Additional comments

L46 – not clear. Is the apple blossom weevil the most important pest? Or is codling moth? Or the two combined? If the paper is about apple blossom weevil, I don’t see the need to mention codling moth at all.

L47 – ‘It is…’ meaning ABW or CM? need to clear up the first sentence to make this make sense

L52 – delete ‘has’ and ‘in the past’

L55 – the most recent reference you have here is 2015 and we’re in 2022. Nothing since then to validate it’s importance?

L58-59 – no recent reports of impact?

L60 – ‘As of today…’ yet the reference is dated 1994. Need to find more recent information for this section.

L65 – rephrase ‘Besides the inconvenient…’ try ‘There are two main monitoring methods: limb jarring (which is inconvenient, Toepfer et al. 1999) and sheltertraps (Hausam….). Sheltertraps are more preferred as they exploit….’

L73-75 – ‘explain why or how flight traps may be a useful component

L75 – ‘insight’

L76 – ‘into the underlying mechanism(s)…’ there could be more than one, there could be just one

L81 – ‘use’ instead of ‘follow’ and ‘Following colonization from adjacent…’

L90-95 – this presumes that the traits the ABW are interested in are heritable and that a breeder can tease out these genes for selection. Might be a better route to use these traits as a screening mechanism for newly developed cultivars. And this depends upon which traits are key to preference.

L140-141 – make sure the connection is made to the timing of selection by ABW

L163 – for host volatiles, the range of these between trees even within a cultivar can be vast. For the 10 trees of 10 different cultivars (L167) how were these chosen? What criteria was used to pick these 10 cultivars? I have concerns with the age range of the trees (1998 – 2008) as host preference can change over time (see Blatt and Hiltz 2021), presumably the VOC profiles will likewise change with time, although I don’t think anyone has looked at this. The use of smaller potted trees (Table S2) for the VOCs may not provide a true match-up with the volatile profiles of the trees from the field (Table S1), particularly as only 5 (2 apple and 3 crab apple) cultivars overlap between field and VOC. I also have concerns that the crab apple trees are significantly larger than the domesticated apple trees and hence the volume of the VOCs would be that much greater – assuming an equivalently larger blossom bud load.

L177-178 – when do they arrive at the trees/blossoms? How long before Stages 56/57? And then L204 says the post-diapause imagines (wouldn’t these be adults?) are on the tree from mid-March to mid-April. It would seem that the selection occurs well before bud stage 56/57, more like at 53/54. Wouldn’t it have made more sense to collect the volatiles from this stage? Once adults have selected a tree do they move between trees?

L215 – cutting of the branches is known to affect host volatiles, not just from the cut surface but through the leaves. Why not collect a second set of VOC from the trees, solvent extract and use these? What confidence do you have that the response is not impacted by any induced volatiles added to the profile?

L223 – sentence is not clear – please rephrase

L235-240 – which cultivars were used in these bioassays? Did they match those preferred and non-preferred from field trials?

L270 – was the ‘solvent delay’ on the MS still used for this method with a thermal desorption of the PDMS twisters (no solvent)?

L395 – ‘stronger’? not the best word here, ‘showed significantly higher infestation by…’. Similar issue on L410, L492, L508

L423-424 – that the M. sylvestris individuals were not located in the botanical garden further impacts this study. The population pressure may be a causal factor in the results obtained. Although mentioned by the authors, the selection of domesticated apple in the botanical garden (near the hedgerows?) could have impacted the results. Need more description of where these cultivars were located within the garden – is distance from the hedgerow a possible factor in the lower levels observed? Were the domesticated trees near the crab apple that was within the garden?

L444 – with no domesticated trees located outside the garden, this is a weak statement

L568 – ‘we’? Piskorski and Dorn 2010? Or the authors of this study? Need to be clear here.

L571-572 – likely because there are overlaps in infestation rates which weren’t examined separately.

L591-592 – again, need to examine crab apples and domesticated apple with different infestation rates. 3/5 cultivars/genotypes had a similar infestation rate.

L616/617 – this is really speculation. Your VOC data wasn’t collected from enough representatives of each apple or crabapple to connect VOCs with infestation rate.

Table 2 – would be interesting to see the comparison of Jonagold and Topaz volatiles compared against Lochau. These all experienced a similar field infestation rate (~7%) – do their profiles show differences? Eimersmuhle and Schlehemuhle both had similar infestation rates (~11%) – were these similar? Grouping the volatiles as done overrides these subtleties, which could provide useful information towards identifying compounds of interest for future management programs.

Reviewer 2 ·

Basic reporting

This manuscript has been written in very well organized way especially the research data material was very much enough and seems to have a good quality of work performed by the authors. The Novelty of this manuscript is that it is among the first and unique study in this research group. Title and abstract is good, while Introduction have required few minor changes. Materials & Methods, Results and discussion is well explained and fully relevant to research question. Whereas, Conclusion needs to little improvement, it must be in one or maximum two paragraphs rather than four.
Few minor changes are recommended below:
Avoid to use old references before 2000. Only use where it seems essential.
Line 45, remove “is”
Line 56, replace “can” with “Have been”
Line 58, replace “can” with “Have been”
Line 89, replace “Since” with “Although”
Line99, remove the word “ovipositing female’, as it seems unfit for this line. Replace this word as “Preference of oviposition in females”
Line 112, remove Comma(,) before and

Experimental design

Experimental design of this research is well described with sufficient replicates. Research question is also relevant to the experiment. Here is need to add few pictures of performing research (insects used in research, performing bioassay etc.) it would enhance the interest and better understanding of the reader toward the novelty of the research.
Line 174, remove “could be” with “and”

Validity of the findings

Results are well described in an appropriate way with all essential tables and figures. statistical analysis also meet the all requirements of the experiment. Overall research data is statistically sound and linked to original research.

Additional comments

It is an interesting study having enough material to accept and also fulfil all requirements according to the umbrella of PeerJ.

Reviewer 3 ·

Basic reporting

no comment

Experimental design

no comment

Validity of the findings

no comment

Additional comments

It is a great pleasure to review the manuscript entitled “Morphological and olfactory tree traits influence the susceptibility and suitability of the apple species Malus domestica and M. sylvestris to the florivorous weevil Anthonomus pomorum (Coleoptera: Curculionidae)” submitted to PeerJ. The topic is very interesting. I believe the present manuscript successfully meets what it intends, with an adequate dataset, analyzed under an effective methodology. The analyzes are overall well-described with suitable methodology and interesting conclusions. In the introduction, materials and methods and results sentences are well-organized and suitable references are provided.

I have some moderate concerns about figure 4 and discussion, and a few minor comments which I list below:

L57-58: replace “In some instances, the” by “The”

L60: replace “As of today, the” by “The”

L60-62: If possible, look for other more recent references to cite here. I recommend authors verify that the following reference cannot be included: https://doi.org/10.3390/insects12121106

Table 1 – This table contains little information that can simply be presented in the text, so I consider it unnecessary.

L187: remove “)”

L311: remove “analysis of deviance;”; Please put “1,18 as subscript.

L327: Replace the words “Chi²” by “X²”
L327: replace “p < 2*10-16” by p < 0.001”

L328-329: I don't understand the presentation of this value of r = 0.397 here. No Mann-Whitney U test performed? I expected here the value of U

L328-329: replace “p < 2*10-16” by p < 0.001”

L333: Please put “1,18 as subscript: F1,18

L334: replace “p < 2*10-16” by p < 0.001”

L341: remove “analysis of deviance;” ; Please put “1,17 as subscript.

L357-358: I suggest the following change to the text: “Most compounds were emitted in higher proportions from blossom buds of M. domestica cultivars compared to M. sylvestris genotypes (Table 2).”

L365: Please cite Table 2 at the end of the paragraph.

L366-368: This whole part of describing the use of PCA should be presented in the methodology, not here.

Figure 4: it is very difficult to observe the elements of this figure since they are very small. Please try to increase the size of the fonts and the different elements shown in the figure.

Discussion: I missed more objective writing as presented in previous sessions (such as methodology and results), perhaps this is due to excessive repetition of results. Furthermore, I believe that the study questions (L147-L158) should be properly developed here.

It bothers me a lot the amount of repeated results that are presented in the eighth paragraph. Lines 468-478. Is it really necessary? I think that only treating (in a comparative way) whether the values were higher or lower than those of a given study would be sufficient here.

Conclusion: This first paragraph should be reformulated and synthesized. L606-607: remove: “, as indicated by significant differences in infestation rates between M. domestica and M. sylvestris”

---

## Round 0.2 · accepted · Accept

Dear Authors
After the revision of the manuscript, reviewers are ready to accept your submission. Thank you for your submission to PeerJ.

Reviewer 1 ·

Basic reporting

The authors have addressed the numerous comments and provided additional information to clarify their choice of trees, timing of VOC collection and have tempered their discussion and conclusions accordingly. The manuscript is greatly improved.

Experimental design

No comment

Validity of the findings

No comment

Additional comments

No comment

Reviewer 2 ·

Basic reporting

Authors have been made all necessary changes in a brief way as all sections have been improved as per suggestions. I have found no ambiguity in this paper according to the language, article structure and description of results. So, I recommend this manuscript for publish in this journal.

Experimental design

Research question and methodology is well defined and all necessary changes have been made to fulfil the PeerJ criteria for publishing the manuscript

Validity of the findings

Current version of the manuscript is well revised by the authors according to the suggestions provided by the reviewers.

Additional comments

No Comment

Reviewer 3 ·

Basic reporting

I am reviewing the updated version of this manuscript. Now, I found an improved manuscript in general. The authors did a good job in order to address my concerns. I believe that the manuscript is now eligible for acceptance.

Experimental design

no comment

Validity of the findings

no comment

Additional comments

no comment